# Ionic Liquid-Based Ultrasonic-Assisted Extraction to Analyze Seven Compounds in *Psoralea Fructus* Coupled with HPLC

**DOI:** 10.3390/molecules24091699

**Published:** 2019-04-30

**Authors:** Mengjun Shi, Juanjuan Zhang, Cunyu Liu, Yiping Cui, Changqin Li, Zhenhua Liu, Wenyi Kang

**Affiliations:** 1National R & D Center for Edible Fungus Processing Technology, Henan University, Kaifeng 475004, China; smj1994@vip.henu.edu.cn (M.S.); liucunyu@vip.henu.edu.cn (C.L.); cuiyiping@vip.henu.edu.cn (Y.C.); LCQ@vip.henu.edu.cn (C.L.); 2Joint International Research Laboratory of Food & Medicine Resource Function, Henan Province, Kaifeng 475004, China; 3Zhengzhou Key Laboratory of Medicinal Resources Research, Huanghe Science and Technology College, Zhengzhou 450063, China; zhangjuan8908@163.com

**Keywords:** *Psoralea Fructus*, ionic liquid (ILs), ultrasonic-assisted method, HPLC

## Abstract

*Psoralea Fructus* is widely used in traditional Chinese medicine (TCM), and the content of psoralen, isopsoralen, neobavaisoflavone, bavachin, psoralidin, isobavachalcone, and bavachinin A is the main quality control index of *Psoralea Fructus* because of its clinical effects. Thus, a fast and environmentally-benign extraction method of seven compounds in *Psoralea Fructus* is necessary. In this work, an ionic liquid-based ultrasonic-assisted method (ILUAE) for the extraction of seven compounds from *Psoralea Fructus* was proposed. Several ILs of different types and parameters, including the concentration of ILs, concentration of ethanol (EtOH), solid–liquid ratio, particle size, ultrasonic time, centrifugal speed, and ultrasonic power, were optimized by the Placket–Burman (PB) design and Box–Behnken response surface analysis. Under this optimal condition, the total extraction yield of the seven compounds in *Psoralea Fructus* was 18.90 mg/g, and significantly greater than the conventional 75% EtOH solvent extraction.

## 1. Introduction

*Psoralea corylifolia* Linn. is an annual plant of the Leguminous family, for which its ripening fruits are used as a kind of traditional Chinese medicine (TCM). It has been used as a herbal remedy to treat psoriasis vulgaris, vitiligo, and alopecia areata, for external application [1,2,3]. At present, 90 compounds have been isolated and identified from *Psoralea Fructus*, including coumarins, flavones, coumarone, meroterpenes, and lipids [4]. Biologically, the major active constituents are coumarins, flavonoids, and meroterpenes, most of which are found in *Psoralea Fructus*. The established bioactivities of these active constituents include antibacterial [5], anti-inflammatory [6], anti-oxidant [7,8], anti-depressant [9,10], anti-tumor and cytotoxicity [11,12], hepatoprotective [13], and estrogenic properties [14,15].

Psoralen and isopsoralen, belonging to furocoumarins, which are important active ingredients in the *Psoralea Fructus* and photosensitizers of natural plants, can promote the synthesis of melanin and treat skin diseases such as vitiligo and psoriasis vulgaris [16,17]. It was reported that psoralidin and neobavaisoflavone exhibited anti-tumor activity, and can treat osteoporosis [18,19]. Moreover, bavachinin A and isobavachalcone belonged to a kind of typical isopentenyl flavonoid, showed an anti-proliferative effect on many kinds of cancer cells [20,21], and a stronger inhibition on *Staphylococcus aureus* (SA) [22,23]. Xin et al. reported that bavachin exhibited significant estrogenic activity [24] and was regarded as the candidate for osteoporosis treatment [25].

In the present study, easy and ecofriendly extraction techniques, such as ionic liquid-based ultrasonic-assisted extraction (ILUAE), ionic liquid-based microwave-assisted extraction (ILMAE), dispersive liquid–liquid microextraction (DLLME), and sugaring-out liquid–liquid extraction (SULLE) coupled with HPLC, are required [26,27,28]. Ionic liquids (ILs), are regarded as molten salts near room temperature, and typically consist of cations and anions. Most importantly, ILs have unique properties, such as a high thermal stability, good solubility, structural modification, and green environmental protection [29,30,31,32,33]. In addition, ILs, as a kind of novel solvent, have been widely used in the extraction and separation of bioactive compounds from TCM, such as flavonoids, coumarins, saponins, and terpenes [34,35,36].

To the best of our knowledge, psoralen, isopsoralen, neobavaisoflavone, bavachin, psoralidin, isobavachalcone, and bavachinin A (Figure 1) in *Psoralea Fructus* extracted with ILs have not yet been reported in the literature. Therefore, the aim of this work was to establish the feasibility method of ILUAE for the analysis of seven ingredients in *Psoralea Fructus*. Moreover, it was reported that ILUAE has been successfully applied in the extraction of the active ingredients, because of the advantages of a reduction in the extraction time and minimum solvent consumption [26,37,38]. Furthermore, Plackett–Burman (PB) design and Box–Behnken response surface design (BBD) were used to optimize the extraction conditions of seven ingredients in *Psoralea Fructus*.

## 2. Result and Discussion

### 2.1. Screening of Ionic Liquid (ILs)

Generally, selecting the appropriate solvent is of great importance in order to obtain a satisfactory extraction efficiency of the target compounds. Firstly, 70% ethanol, methanol (MeOH), acetonitrile, and water, as three kinds of common solvent for the extraction of TCM, were compared in our study [39,40]. The results showed that water was not suitable to be used to extract flavonoids and coumarin from *Psoralea Fructus*. In Figure 2a, 70% EtOH was the best solvent for extracting the target analytes, which was significantly higher than MeOH and acetonitrile (*p* < 0.001). Therefore, 70% EtOH was selected as the solvent in the following studies.

Based on previous experiments, ILs were added to increase the extraction yields of the target compounds, which were greatly affected by the chemical structures of the ILs and its physicochemical properties [41,42]. The effects of four kind of ILs with 70% EtOH on the target analytes were compared, which indicated that the extraction yields of using the ILs with 70% EtOH were higher than solely 70% EtOH (Figure 2b; *p* < 0.01). Moreover, the results showed that the highest extraction rate of the target analytes was obtained by using [HMIM]PF_6_/70% EtOH, and the extraction rate of the target analytes by [BMIM]PF_6_/70% EtOH was significantly lower than that of the highest (*p* < 0.05), which may be related to the strong interactions of π–π, n–π, and hydrogen bond between the target analytes and ILs [43]. In view of the facts that [HMIM]PF_6_/70% EtOH exhibits the highest extraction efficiency, it was thus selected for the subsequent experiments.

### 2.2. Single Factor Experiments

#### 2.2.1. Selection of Concentration of ILs

The optimum ILs concentration for the maximum extraction of seven compounds in *Psoralea Fructus* was carried out with varying concentrations, from 0.2 to 1.6 M (Figure 3a). The results showed that the extraction yields increased remarkably with increasing [HMIM]PF_6_ from 0.2 to 0.6 M (*p* < 0.05), and achieved the highest at 0.6 M, and then, the extraction efficiency decreased. This may be because along with the viscosity increased and the diffusivity of the ILs solution decreased, and the ILs solution was difficult to enter to the structure of the sample, and extraction of the target compounds was decreased [44]. Thus, 0.6 M [HMIM]PF_6_ was considered as the optimum concentration, and was chosen for further assays.

#### 2.2.2. Selection of Concentration of EtOH

Based on the previous experiments, 70% EtOH was considered the optimum solution for dissolved ILs. It was also reported that the volume concentration of EtOH was an important parameter affecting the extraction efficiency [45], and a high-volume concentration of EtOH was good for the extraction of the target analytes [46,47]. In Figure 3b, the extraction yield significantly increased with increasing the volume concentration of EtOH (*p* < 0.01), and remained basically unchanged when it increased to 80%. Hence, 80% EtOH was selected as the optimum extraction parameter for further assays.

#### 2.2.3. Selection of Solid–Liquid Ratio

Different solid–liquid ratios of 1:5, 1:10, 1:20, 1:50, 1:80, and 1:120 (g/mL) were used for the experiment. In Figure 3c, the extraction yield tended to markedly increase with the risen proportion of liquid (*p* < 0.05), and achieved the highest at the solid–liquid ratio of 1:80. However, when the solid–liquid ratio was too high, the partition equilibrium of the target analytes in the ILs had not yet been reached. The main reasons may be that when the solid–liquid ratio increased gradually, it led to a difference between the intracellular and extracellular concentration, which become the driving force in the extraction process [48]. So, 1:80 solid–liquid ratios were selected for further assays.

#### 2.2.4. Selection of Concentration of Particle Size

In Figure 3d, with the increase of the grinding mesh, the extraction yields of seven compounds from *Psoralea Fructus* at 20 and 50 meshes were significantly higher than the other meshes (*p* < 0.05), and reached the maximum at 50 meshes. This can be explained because the ILs have a characteristic of viscidity, which leads to forming a cluster of the sample, hindering the extraction effect [49]. Therefore, 50 meshes were selected for the subsequent experiments.

#### 2.2.5. Selection of Concentration of Ultrasonic Time

To know the optimum time for the better extraction concentration of the target compounds, different extraction times from 10 to 60 min were investigated. In Figure 3e, the yield of seven compounds from *Psoralea Fructus* was increased markedly from 10 to 20 min (*p* < 0.01), but no longer increased after 20 min (*p* > 0.01). An increase in the time of the extraction may result in structural changes of ILs [50], and decrease in the extraction yield. Hence, 20 min was selected for the subsequent experiments.

#### 2.2.6. Selection of Concentration of Centrifugal Speed

An appropriate centrifugal speed can not only save time and reduce energy consumption, but also ensure that the extraction process is fully carried out. On the basis of the above optimized conditions, different centrifugal speeds ranging from 3000 to 9000 r/min were chosen to evaluate the effect of the centrifugal speed on the extraction yield. In Figure 3f, the extraction rate reached the maximum at 4000 r/min, which was significantly higher than that of 3000 and 7000 r/min, and there was no significant difference with 5000, 6000, and 7000 r/min (*p* > 0.05). Thus, the centrifugal speed of 4000 r/min was selected as the optimal centrifugal speed for the further experiments.

#### 2.2.7. Selection of Concentration of Ultrasonic Power

For knowing the impact of the ultrasonic power on the yield of seven compounds from *Psoralea Fructus*, different ultrasonic powers were used (120, 200, 300, 400, and 500 W). When the ultrasonic power was 300 W, the extraction yield of the target compounds was the highest (Figure 3g). It was found that the extraction yield of the target compounds apparently increased from 120 to 200 W (*p* < 0.05), and there was no significant difference from 200 to 500 W (*p* > 0.05), which may be the reason that with the increasing power, the effects of the ultrasound induce a greater penetration of the solvent into the sample, and improve the extraction rate [51]. However, excessive ultrasound power may destroy the cells of the herb medicine, and affect the extraction yield. Thus, the ultrasonic power of 300 W was selected as the ultrasonic power for the further experiments.

### 2.3. Screening of Significant Factors Using a Plackett–Burman (PB) Design

The PB design was employed to investigate the effect the ILUAE parameters on the active ingredients of *Psoralea Fructus*. Here, seven parameters (A: concentration of ILs; B: concentration of EtOH; C: solid–liquid ratio; D: particle size; E: ultrasonic time; F: centrifugal speed; G: ultrasonic power) were considered as the independent variables that may affect the yield of the target analytes in the extraction process, and to screen some significant factors for further optimization. In Table 1, seven factors affecting the designated responses (total extraction yield Y) were tested by 12 experimental runs using PB design. Then, the PB design was analyzed by regression analysis and the analysis of variance (ANOVA) (Table 2). The results indicated that B, C, Dl and F were significant for the yields, whereas the other parameters contributed non-significantly.

Therefore, it was found that among the seven parameters tested, the concentration of EtOH (B), solid–liquid ratio (C), particle size (D), and centrifugal speed (F) were chosen as the positive parameters for further optimization, whereas the other three parameters, including the concentration of ILs (A), ultrasonic timer (E), and ultrasonic power (G), showed negative effects on the extraction process. In addition, the liquid–solid ratio was also regarded as the significant parameter in other extraction processes [52,53].

The model equations developed by the PB design for the total extraction yield was as follows: *Y* = 15.53 + 1.54A − 0.99B − 1.03C − 2.67D − 0.35E − 0.59F − 0.57G. The value of the adjusted determination coefficients (R^2^adj) was 0.9230, close to the correlation coefficient (R^2^), which suggested that this model could be explained better.

### 2.4. Optimization of Parameters by the Box–Behnken Design (BBD)

Based on the results obtained from the analysis of the PB design, the concentration of EtOH, solid–liquid ratio, particle size, and ultrasonic time were performed significantly, and optimized for the subsequent studies by BBD, which keeps the other three factors—concentration of ILs, centrifugal speed, and ultrasonic power—consistent with the level of 0.6 M, 20 min, and 300 W, respectively. Three levels for each parameter are usually required, however, there were run orders where all of the variables at their lower and higher areas were not encompassed, making an experimental design more economical [54]. In this study, the total extraction yields of the compounds in *Psoralea Fructus* were taken as the response values (*Y*), the concentration of EtOH (*X*_1_), solid–liquid ratio(*X*_2_), particle size (*X*_3_), and centrifugal speed (*X*_4_), four factors by four-factors-three-levels of response surface analysis. The experimental factors and level design for the BBD analysis are shown in Table 3. In order to study the effect of the various parameters and their interactions on the extraction process, the data in Table 4 are fitted with multiple regression by Design Expert 8.0.6 software, in order to obtain the following quadratic multiple regression equation: *Y* = 13.82 − 0.59*X*_1_ + 0.76 *X*_2_ − 1.41 *X*_3_ − 0.077*X*_4_ − 0.61 X_1_*X*_2_ + 0.51 *X*_1_*X*_3_ + 0.25*X_1_X_4_* − 1.18*X_2_X_3_* + 0.30*X*_2_*X*_4_ + 0.020*X*_3_*X*_4_ − 1.00*X*_1_^2^ − 0.54*X*_2_^2^ + 1.80*X*_3_^2^ + 0.16*X*_4_^2^ (3).

The ANOVA of the response surface quadratic model is shown in Table 5. The results indicate that the model was significant and adequate for the reasonable prediction of the total extraction yields of compounds in *Psoralea Fructus,* within the variable range employed, as evidenced by the *F*-value (4.82) and the low probability values (*p* < 0.01). The coefficient of determination (R^2^) for the quadratic regression model was 0.8283, which was higher than 0.7, indicating that the model was workable for using in the experiment. Therefore, this model can be considered as better fitting [55]. From the results shown in Table 5, it is indicated that the most important factor affecting the extraction efficiency of the target components is the particle size, which is consistent with the screening results of the PB test.

In order to investigate and visualize the impacts of the factors and the total extraction yield, the response surface maps are used for the three-dimensional relationship between the independent variables, and the response value are shown in Figure 4. The steeper the shape of the response surface maps is, the more obvious the interaction is, for which its trend reflects the trend of the response value changing with various factors. In Figure 4, the interaction of concentration of EtOH (*X*_1_) and the solid–liquid ratio (*X*_2_) are the strongest, while that of the solid–liquid ratio (*X*_2_) and centrifugal speed (*X*_4_) are the weakest. The interactions between the concentration of EtOH (*X*_1_) and the solid–liquid ratio (*X*_2_) are presented in Figure 4a. With the increase in concentration of EtOH (*X*_1_), the extraction efficiency of the target analytes firstly increased, and then decreased, which was consistent with the results of a single factor. With the increase of the solid–liquid ratio (*X*_2_), the extraction efficiency of the target analytes increased in the beginning, and then became lower, which was consistent with the results of a single factor.

According to the response surface analysis, the optimum extraction parameters adjusted by the integer of *Psoralea Fructus* were obtained as follows: the concentration of ILs was 0.6 M, concentration of EtOH was 75%, the solid–liquid ratio was 1:40, the particle size was 20 meshes, the ultrasonic time was 20 min, the centrifugal speed was 3000 r/min, and the ultrasonic power was 300 W. The validity of the response surface analysis method was evaluated by seven compounds in *Psoralea Fructus* for three replicate verification experiments under the optimum conditions. The actual yield of the seven compounds in *Psoralea Fructus* was 18.90 mg/g, which was similar to the predicted values (19.34 mg/g), which indicated that the BBD was reliable.

### 2.5. Comparison with Conventional Solvents Extraction

To evaluate the extraction efficiency of the different methods, the proposed ILUAE approach under optimal conditions was compared with the traditional solvent–EtOH extraction. Under the optimal conditions of the [HMIM]PF_6_/75%EtOH extraction, the total extraction yield of the seven compounds in *Psoralea Fructus* was 18.90 ± 0.67 mg/g (*n* = 3), while the total extraction yield of the seven compounds in *Psoralea Fructus* obtained using traditional solvent–75% EtOH was 14.72 ± 0.25 mg/g (*n* = 3). Obviously, the ILUAE significantly increased the extraction rates. The results indicate that the extraction process was optimized by PB design and BBD.

### 2.6. Method Validation

Calibration curves were constructed by plotting the peak area versus the extraction yield of the target analytes, and was obtained by the analysis of five different concentration levels of the standard solutions in triplicate. The linear regression equations, linear ranges, correlation coefficients, linearity, limit of detection (LODs; based on a signal-to-noise ratio of 3; S/N = 3), and the limit of quantification (LOQs; based on a signal-to-noise ratio of 10; S/N = 10) are listed in Table 6.

The precision was measured by six consecutive injections of standard solution, and the result showed relative standard deviation (RSD) values of 0.15%, 0.10%, 1.20%, 0.25%, 2.67%, 3.44%, and 0.35%, respectively, indicating that the precision of the instrument was good and could accurately reflect the amount of the substance. Moreover, the recovery tests were measured using the standard-addition method at six different concentration levels. A good recovery, ranging from 86.72% to 103.71%, was obtained, and the RSD values were lower than 3.81%, which indicated that the above method was credible. The stability was investigated by the determination of six extracts of the samples under the optimized ILUAE procedure, and the RSD values of the seven compounds were 2.07%, 3.88%, 3.80%, 3.82%, 3.51%, 2.10%, and 2.91%, respectively.

## 3. Materials and Methods

### 3.1. Materials and Reagent

*Psoralea Fructus* were obtained from the Pharmacy of Zhang Zhongjing, in July 2017, and identified by Professor Changqin Li. The voucher specimen was deposited in the National R&D Center for Edible Fungus Processing Technology.

Acetonitrile (chromatographic grade ≥99.9) and methanol (chromatographic grade ≥99.9) were purchased from the Xilong Scientific Factory (Guangdong, China). The pure water was purchased from Hangzhou WahahaBaili Food Co., Ltd, (Zhejiang, China). The 1-butyl-3-methylimidazolium tetrafuoroborate ([BMIM]BF_4_; ≥98), 1-butyl-3-methylimidazole bromide ([BMIM]Br; ≥98) and 1-butyl-3-methylimidazolium hexafuorophosphate ([BMIM]PF_6_; ≥98) were obtained from a limited partnership Merck (Darmstadt, German). The 1-hexyl-3-methylimidazolium hexafuorophosphate ([HMIM]PF_6_, ≥98) was purchased from Thermo Fisher Scientifc (Rockville, MD, USA). The psoralen, isopsoralen, neobavaisoflavone, bavachin, psoralidin, isobavachalcone, and bavachinin with a purity greater than 98% were isolated in our previous chemical research [22].

### 3.2. Instrument and Chromatographic Conditions

A LC-20AT high-performance liquid chromatography system (Shimadzu, Kyoto, Japan) equipped with a degasser, a quaternary gradient low pressure pump, the CTO-20A column oven, a SPD-M20AUV-detector, and a SIL-20 auto sampler was used. TGL-16 type high-speed centrifuge was obtained from the Jiangsu Jintan Zhongda instrument factory (Jiangsu, China). The AB135-S 1/10 million electronic balance was purchased from Mettler Toledo Instruments Co., Ltd (Shanghai, China).

The chromatographic separations of the target analytes were performed on a RP-18 endcapped column (4.6 mm × 250 mm; 5 µm). The mobile phase was a mixture of acetonitrile (A) and water (B). The gradient elution steps were set as follows: 0~30 min, 25%~45% A; 30~60 min, 45%~80% A; 60~70 min, and 80% A. The flow rate of the mobile phase was maintained at 0.8 mL/min. The column temperature was controlled at 30 °C. The UV detection wavelength was set at 246 nm. The HPLC chromatograms of the standard solution and the sample extract are shown in Figure 5.

### 3.3. Preparation of the Standard Solution

Seven standard solutions of psoralen, isopsoralen, neobavaisoflavone, bavachin, psoralidin, isobavachalcone, and bavachinin were prepared in methanol, at a concentration of 107.5, 111.9, 34.42, 15.36, 51.52, 18.84, and 69.60 mg/mL, respectively, and were stored at 4 °C.

### 3.4. Screening of ILs

The powder of *Psoralea Fructus* (50 mg; 20 meshes) and 2 mL of aqueous solutions of the different ILs were placed in a volumetric flask. After ultrasonic extraction for 20 min at an ultrasonic power of 300 W, the mixture was centrifuged at 3000 r/min for 5 min. Before the HPLC analysis, the supernatant was filtered with a 0.22 μm organic microporous membrane. Each sample was performed in triplicate. Among the seven compounds, there were three ingredients of coumarin, including psoralen, isopsoralen, and psoralidin, and four ingredients of flavone, including neobavaisoflavone, bavachin, isobavachalcone, and bavachinin, for which the weight coefficients of the two indexes were 0.6 and 0.4, respectively. The total extraction yields (mg/g) of the target analytes was determined according to the following formula (1): (1)$$\mathrm{Total}\;\mathrm{extraction}\;\mathrm{yield}\left(\mathrm{mg}/g\right)\;=\frac{\mathrm{mean}\;\mathrm{mass}\;\mathrm{of}\;\mathrm{coumarin}\;\mathrm{in}\;\mathrm{samples}\;(\mathrm{mg})\times \;0.6\;+\;\mathrm{mean}\;\mathrm{mass}\;\mathrm{of}\;\mathrm{flavone}\;\mathrm{in}\;\mathrm{samples}\;(\mathrm{mg})\;\times \;0.4}{\mathrm{mean}\;\mathrm{mass}\;\mathrm{of}\;\mathrm{samples}\;(g)}$$

### 3.5. Optimization of Parameters for ILUAE by the Plackett–Burman (PB) Design and Response Surface Methodology (RSM)

The single factor experiments were performed in several combinations, with seven parameters, including different IL concentrations (0.2–1.6 M), concentration of ethanol (40%–100%), liquid–solid ratios (5–120 g/mL), particle size (20–70 mesh), ultrasonic time (10–90 min), centrifugation speed (3000–9000 r/min), and ultrasonic power (120–500 W) in the ultrasonic extractor. The significant factors were chosen by the PB design using Design-Expert 8.0.6 Trail software (Stat-Ease, Minneapolis, MN, USA). Then, a Box–Behnken design (BBD) was applied for the RSM optimization using Design-Expert 8.0.6 Trail software.

### 3.6. Conventional Solvent Procedures

The optimum conditions (IL excluded) were used for measuring for the following experiments. All of the experiments were performed in triplicate. The extracts were filtered through a microporous membrane for the HPLC analysis.

## 4. Conclusions

In this work, an ILUAE method was proposed to extract psoralen, isopsoralen, neobavaisoflavone, bavachin, psoralidin, isobavachalcone, and bavachinin A from *Psoralea Fructus*. Compared with traditional methods, the extraction yields of the present approach may increase by 28.40%, and the ILs used as a solvent can be recycled by some methods, such as membrane filtration and liquid–liquid extraction [56,57]. Thus, the IL-based ultrasonic-assisted extraction is a simple, rapid, and highly efficient extraction technique for the seven compounds—psoralen, isopsoralen, neobavaisoflavone, bavachin, psoralidin, isobavachalcone, and bavachinin A—from *Psoralea Fructus*.

## Figures and Tables

**Figure 1 molecules-24-01699-f001:**
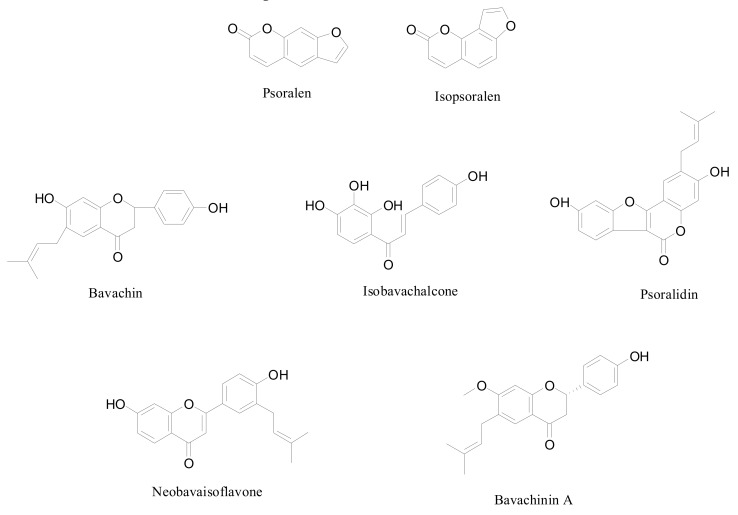
Chemical structures of seven compounds from *Psoralea Fructus*.

**Figure 2 molecules-24-01699-f002:**
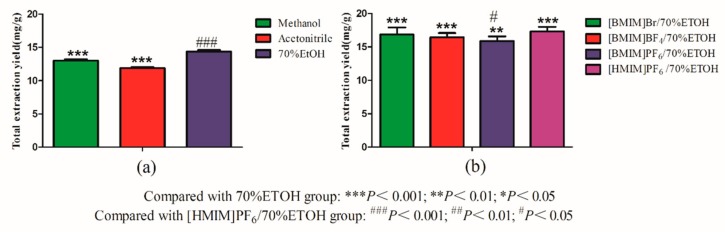
Effect of (a) extraction solvents and (**b**) ionic liquids (ILs) (*n* = 3).

**Figure 3 molecules-24-01699-f003:**
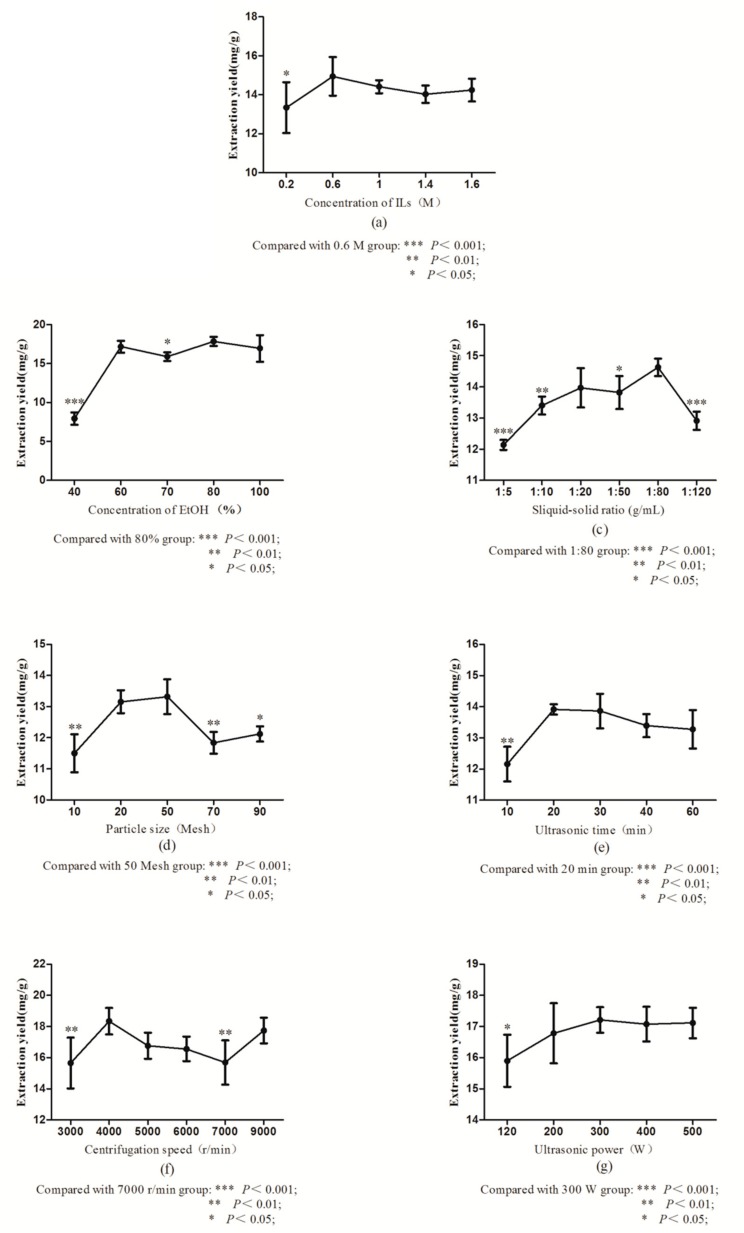
Effect of concentration of (**a**) ILs, (**b**) concentration of EtOH, (**c**) liquid–solid ratios, (**d**) particle size, (**e**) ultrasonic times, (**f**) centrifugation speed, and (**g**) ultrasonic power on seven compounds in *Psoralea Fructus* (*n* = 3).

**Figure 4 molecules-24-01699-f004:**
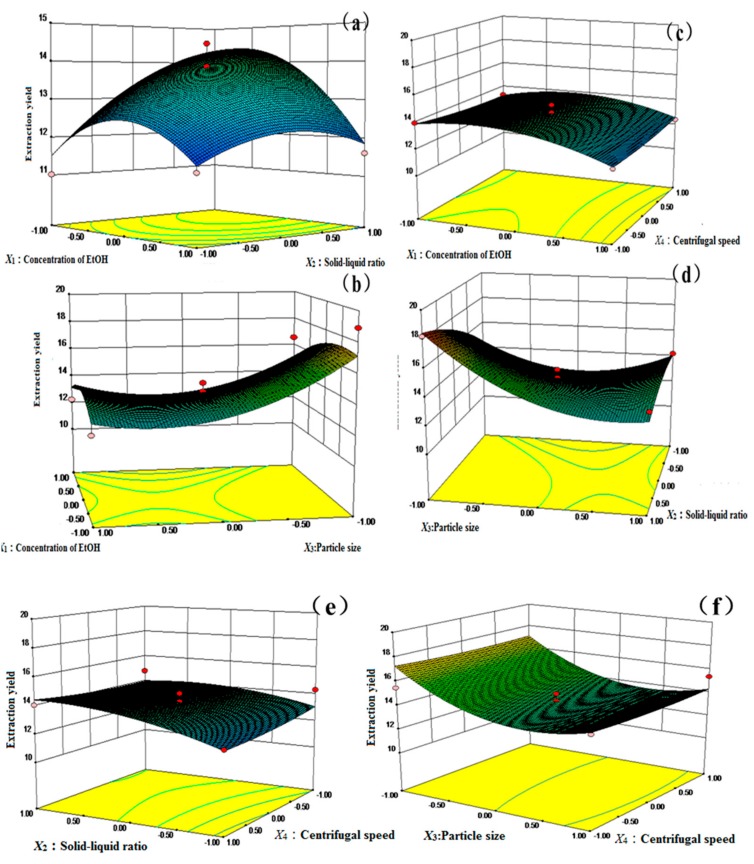
Response surface plots showing the effects of variables on the total extraction yield of seven compounds from *Psoralea Fructus*; (**a**) X_1_ and X_2_; (**b**) X_1_ and X_3_; (**c**) X_1_ and X_4_; (**d**) X_2_ and X_3_; (**e)** X_2_ and X_4_; (**f**) X_3_ and X_4_.

**Figure 5 molecules-24-01699-f005:**
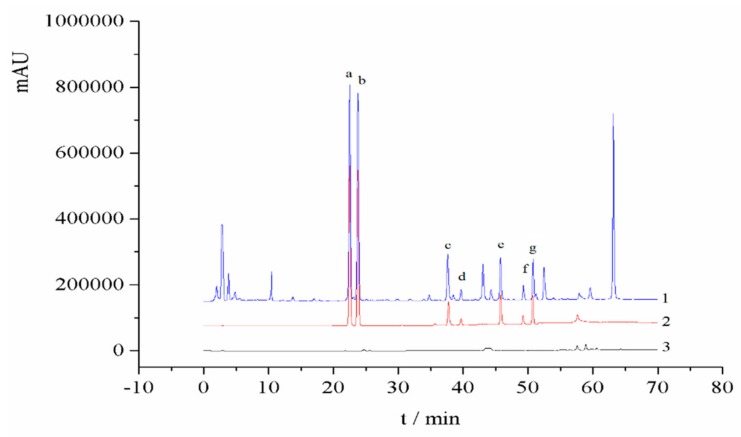
HPLC chromatograms of the (1) sample solution, (2) standard solution, (3) and blank solvent, namely: (**a**) psoralen, (**b**) isopsoralen, (**c**) neobavaisoflavone, (**d**) bavachin, (**e**) psoralidin, (**f**) isobavachalcone, and (**g**) bavachinin.

**Table 1 molecules-24-01699-t001:** Factors (in coded levels) of the Plackett–Burman (PB) design with total extraction yield (mg/g) as a response.

No.	A (M/L)	B (%)	C (g/mL)	D (Mesh)	E (min)	F (r/min)	G (W)	Total Extraction Yield (mg/g)
1	1(1)	−1(80%)	−1(1:120)	−1(20)	1(30)	−1(3000)	1(500)	17.85
2	−1(0.2)	−1	−1	−1	−1(10)	−1	−1(200)	17.10
3	−1	−1	−1	1(70)	−1	1 (5000)	1	13.84
4	−1	1(100%)	−1	1	1	−1	1	11.67
5	1	1	1(1:50)	−1	−1	−1	1	17.00
6	1	1	−1	1	1	1	−1	13.38
7	−1	1	1	−1	1	1	1	14.22
8	1	−1	1	1	1	−1	−1	13.88
9	−1	−1	1	−1	1	1	−1	19.87
10	1	1	−1	−1	−1	1	−1	21.78
11	1	−1	1	1	−1	1	1	14.18
12	−1	1	1	1	−1	−1	−1	11.53

**Table 2 molecules-24-01699-t002:** Analysis of variance (ANOVA) and regression analysis of PB design for the prediction of significant extraction variables.

Source	Sum of Squares	Degree of Freedom	Mean Square	*F*-Value	*p*-Value	Inference
Model	112.46	10	11.25	935.34	0.0254	Significance
Residua	0.012	1	0.012			
Cor total	112.47	11				
	R^2^ = 0.9999; R_agj_^2^ = 0.9988					
**Term**	**Mean Square**	*F* **-Value**	*p* **-Value**			
A	11.25	935.34	0.061			
B	18.95	1576.04	0.024	Significance		
C	8.49	705.83	0.0276	Significance		
D	6.4	532.15	0.0116	Significance		
E	36.14	3006.1	0.0706			
F	0.97	80.74	0.0418	Significance		
G	2.79	231.66	0.066			

**Table 3 molecules-24-01699-t003:** Box–Behnken design (BBD) for the independent variables and corresponding response values.

Level	*X*_1_ (%)	*X*_2_ (g/mL)	*X*_3_ (Mesh)	*X*_4_ (r/min)
−1	60	1:120	20	3000
0	80	1:80	40	4000
1	100	1:40	60	5000

**Table 4 molecules-24-01699-t004:** Experimental design and results of BBD.

Run Order	X_1_	X_2_	X_3_	X_4_	Extraction Yield (mg/g)		Total Extraction Yield (mg/g)
					Coumarin	Flavone	
1	0	1	0	1	16.03	11.07	14.04
2	1	0	0	−1	13.16	10.47	12.09
3	0	0	0	0	15.69	10.56	13.64
4	1	0	0	1	13.33	11.00	12.39
5	1	0	1	0	13.84	9.82	12.23
6	0	−1	1	0	16.59	10.66	14.22
7	−1	−1	0	0	13.05	7.96	11.01
8	0	0	1	1	17.93	11.56	15.38
9	0	0	0	0	16.03	9.87	13.57
10	−1	1	0	0	16.73	9.05	13.66
11	0	−1	0	−1	16.46	11.09	14.31
12	−1	0	−1	0	22.95	12.84	18.91
13	0	−1	−1	0	17.44	8.29	13.78
14	0	0	0	0	16.74	11.01	14.45
15	0	1	1	0	16.61	9.87	13.91
16	0	0	−1	−1	18.79	10.53	15.48
17	−1	0	1	0	14.89	9.00	12.54
18	0	1	0	−1	17.02	11.46	14.79
19	0	0	0	0	16.17	9.78	13.61
20	1	0	−1	0	19.35	12.36	16.55
21	−1	0	0	1	15.55	9.87	13.28
22	0	0	0	0	16.19	10.34	13.85
23	−1	0	0	−1	16.22	10.59	13.97
24	0	−1	0	1	14.52	9.11	12.35
25	0	1	−1	0	21.32	13.54	18.21
26	0	0	−1	1	20.23	10.97	16.52
27	0	0	1	−1	17.09	10.02	14.26
28	1	1	0	0	12.55	10.16	11.60
29	1	−1	0	0	12.56	9.61	11.38

**Table 5 molecules-24-01699-t005:** ANOVA for the fitted quadratic polynomial model for the optimization of extraction parameters.

Source	Sum of Squares	df	Mean Square	*F* Value	*p* Value
Model	80.22	14	5.73	4.82	0.0029 **
*X* _1_	4.22	1	4.22	3.55	0.0804
*X* _2_	7.01	1	7.01	5.9	0.0292
*X* _3_	23.83	1	23.83	20.06	0.0005
*X* _4_	0.071	1	0.071	0.06	0.8098
*X* _1_ *X* _2_	1.48	1	1.48	1.24	0.2834
*X* _1_ *X* _3_	1.05	1	1.05	0.88	0.3637
*X* _1_ *X* _4_	0.25	1	0.25	0.21	0.6535
*X* _2_ *X* _3_	5.61	1	5.61	4.72	0.0474
*X* _2_ *X* _4_	0.36	1	0.36	0.31	0.5889
*X* _3_ *X* _4_	0.0016	1	0.0016	0.0014	0.9712
*X* _1_ ^2^	6.47	1	6.47	5.45	0.035
*X* _2_ ^2^	1.89	1	1.89	1.59	0.2274
*X* _3_ ^2^	21.06	1	21.06	17.73	0.0009
*X* _4_ ^2^	0.17	1	0.17	0.14	0.7113
Residual	16.63	14	1.19		
Lack of fit	16.1	10	1.61	12.09	0.0141 *
Pure error	0.53	4	0.13		
Cor total	96.85	28			

** Highly significant (*p* < 0.01); * Significant (*p* < 0.05).

**Table 6 molecules-24-01699-t006:** Regression equations, linear ranges, limit of detection (LODs), and limit of quantification (LOQs) of the analytes.

Analyte	Regression Equation	Linear Range (μg)	Correlation Coefficient	LOD (ng)	LOQ (ng)
Psoralen	*Y* = 6467788*X* + 592182	0.2150~9.677	0.9998	0.2150	0.4301
Isopsoralen	*Y* = 8303251*X* − 504662	0.2238~10.07	0.9999	0.2238	0.4476
Neobavaisoflavone	*Y* = 4085132*X* + 42249.1	0.06885~3.098	0.9999	0.4131	1.3770
Bavachin	*Y* = 3233926*X −* 86602.6	0.03072~1.382	0.9974	0.6144	1.5360
Psoralidin	*Y* = 3225382*X* + 2381.59	0.1030~4.637	0.9999	0.3091	2.0608
Isobavachalcone	*Y* = 2246978*X* + 579.852	0.03768~1.696	0.9999	0.4522	1.8840
Bavachinin A	*Y* = 1943405*X* + 66464.4	0.1392~6.264	0.9996	0.2784	2.7840

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
