# Peer review of "Ionic Liquid-Based Ultrasonic-Assisted Extraction to Analyze Seven Compounds in Psoralea Fructus Coupled with HPLC"

_molecules, 2019, doi:10.3390/molecules24091699_

Round 1

Reviewer 1 Report

The manuscript by Shi et al focuses on the analysis of compounds in P. Fructus using IL-based ultrasonic-assisted extraction technique. Overall, the manuscript is described well. However, the following suggestions are recommended before accepting the manuscript.

Line 48: Please provide a complete definition of ILs. As the authors described it is novel but well investigated.

Line 51: Briefly described the application of ILs in other extraction techniques.

Line 64: Optimization of EtOH as a solvent is first described before IL selection. So which IL was used in this experiment. It was not clear and straight forward. Why authors did not use pure ethanol whereas for MeOH and Acetonitrile is in 100%?

Line 93: 80% vs 70% EtOH difference in the single factor optimization study. Experiments and writing would have more organized than this.

Line 97: Is it 3(d) or 3(c)?

Line 229: For method validation, did authors try real samples? It is my understanding that all experiments were conducted with previously extracted analytes. In such a situation how do you justify the matrix effect for the extraction efficiency?

Line 244: MeOH and EtOH chemical purity and from where they were purchased are not described. The purity of ILs purchased is not described.

Line 256: Please provide the citation.

Line319: Please check reference formatting

Author Response

Response to reviewer 1: 1. Line 48: Please provide a complete definition of ILs. As the authors described it is novel but well investigated. Reply: Thanks for your comments. We have added a complete definition of ILs in the manuscript and marked it in red from Line 51 to Line 53. 2. Line 51: Briefly described the application of ILs in other extraction techniques. Reply: Thanks for your comments. We have described the application of ILs in other extraction techniques in the manuscript and marked it in red from Line 53 to Line 56. 3. Line 64: Optimization of EtOH as a solvent is first described before IL selection. So which IL was used in this experiment. It was not clear and straight forward. Why authors did not use pure ethanol whereas for MeOH and Acetonitrile is in 100%? Reply: Thanks for your suggestion. I am sorry that the ambiguity of my expression caused your confusion. I have revised them and marked in red in section 2.1. According the references, 70% ethanol, pure MeOH and acetonitrile were used as common organic reagent to extract the bioactive compounds in TCM. In our study, three kinds of solvents (70% Ethanol, pure MeOH and acetonitrile) were chose to look for the best extracting solvent for the target compounds. And, based on the these experiments, ILs was added to further improvement of extraction rate and the different concentrations of EtOH was selected in the following experiment. 4. Line 93: 80% vs 70% EtOH difference in the single factor optimization study. Experiments and writing would have more organized than this. Reply: Thanks for your comments. In our study, 70% EtOH, chosen as one kind of common solvent, was used to determine the best extraction solvent. And based on this condition, the concentration of EtOH range from 40 to 100% mixed with ILs (optimized by previous experiment) was selected as an important parameter to optimize the extraction condition. 5. Line 97: Is it 3(d) or 3(c)? Reply: Thanks for your comments. After checking the manuscript, 3(c) is right in Line 105 and I have revised it. 6. Line 229: For method validation, did authors try real samples? It is my understanding that all experiments were conducted with previously extracted analytes. In such a situation how do you justify the matrix effect for the extraction efficiency? Reply: Thanks for your suggestion. I have revised the manuscript in section of 2.6. For method validation, some parameters such as linearity, LODs, and the LOQs, precision, and recovery of the proposed method were investigated with previously extracted analytes. And, reproducibility of methods was conducted with afterwards extracted analytes. 7. Line 244: MeOH and EtOH chemical purity and from where they were purchased are not described. The purity of ILs purchased is not described. Reply: Thanks for your comments. We have added the detailed information of MeOH, EtOH, and ILs including the place of production and purity from Line 254 to 260. 8. Line 256: Please provide the citation. Reply: Thanks for your suggestion. We have added the citation in Line 262. 9. Line 319: Please check reference formatting. Reply: Thanks for your comments. We have checked the reference formatting.

Reviewer 2 Report

I suggest to the Authors to add in the introduction section the following references related to the application of NADES (a specific type of IL):

Food and Chemical Toxicology (ISSN: 0278-6915), 119, 189-198, 2018. 

Molecules (ISSN 1420-3049), 24(7), 1226, 1-16, 2019

Additionally, i suggest to improve the figures 2 and 3 discussion and to report the validation of the HPLC method (or at least the reference)

Author Response

Response to reviewer 2: 1. I suggest to the Authors to add in the introduction section the following references related to the application of NADES (a specific type of IL): Food and Chemical Toxicology (ISSN: 0278-6915), 119, 189-198, 2018. Molecules (ISSN 1420-3049), 24(7), 1226, 1-16, 2019. Additionally, I suggest to improve the figures 2 and 3 discussion and to report the validation of the HPLC method (or at least the reference) Reply: Thanks for your suggestion. We have added in the introduction section the references related to the application of a specific type of ILs in Line 50. Additionally, we have improved discussions about figures 2 and 3. And, the method validation of HPLC have been revised in section 2.6 of the manuscript.

Reviewer 3 Report

The manuscript “Ionic liquid-based ultrasonic-assisted extraction to analyze seven compounds in Psoralea Fructus coupled with HPLC” by M. Shi and colleagues investigated the use of ILUAE to extract several biomolecules with potential pharmacological activity from Psorales Fructus. The article is well-organized although it needs several English and editing corrections. Also, the bibliography needs to be properly formatted. On the other hand, in my opinion, the criteria use for selecting the most important variables are questionable and the discussed differences are not so clear by visual inspection. For example, in Figure 2 p-values must be calculated and reported to show the statistical significance of the results. Similarly, in Section 2.2.1 “the extraction yields increased remarkably with increasing [HMIM]PF6 in the range 0.2 to 0.6 M and was achieved at the highest at 0.6 M, and then, extraction efficiency decreased”. However, a statistical test is not reported and, by looking at the error bars in figure (3a), a similar trend is quite speculative and no apparent difference is observed in the 0.6-1.6 M interval. Similar considerations apply to the other graphs, i.e. all missing a statistical test (Figures 2 and 3).

Other minor observations:

The 2 studies in Fig. 2 (solvent and ILs) should be separated (i.e. Fig2a, Fig 2b). The actual representation is misleading.

Please add the standard deviation to the obtained extraction values (lines 225-226).

Line 233: I suppose the authors refer to the concentration of the analyte (not the mass).

Author Response

Response to reviewer 3: 1. The manuscript “Ionic liquid-based ultrasonic-assisted extraction to analyze seven compounds in Psoralea Fructus coupled with HPLC” by M. Shi and colleagues investigated the use of ILUAE to extract several biomolecules with potential pharmacological activity from Psorales Fructus. The article is well-organized although it needs several English and editing corrections. Also, the bibliography needs to be properly formatted. On the other hand, in my opinion, the criteria use for selecting the most important variables are questionable and the discussed differences are not so clear by visual inspection. For example, in Figure 2 p-values must be calculated and reported to show the statistical significance of the results. Similarly, in Section 2.2.1 “the extraction yields increased remarkably with increasing [HMIM]PF6 in the range 0.2 to 0.6 M and was achieved at the highest at 0.6 M, and then, extraction efficiency decreased”. However, a statistical test is not reported and, by looking at the error bars in figure (3a), a similar trend is quite speculative and no apparent difference is observed in the 0.6-1.6 M interval. Similar considerations apply to the other graphs, i.e. all missing a statistical test (Figures 2 and 3). Reply: Thanks for your comments. First, the bibliographies have been properly formatted. Second, the variables we selected by One-way ANOVA and the corresponding modifications we did were marked in red. Also, we made English editing carefully. 2. The 2 studies in Fig. 2 (solvent and ILs) should be separated (i.e. Fig2a, Fig 2b). The actual representation is misleading. Reply: Thanks for your suggestion. We have separated the Fig. 2 as Fig.2a (solvent) and Fig.2b (ILs). 3. Please add the standard deviation to the obtained extraction values (lines 225-226). Reply: Thanks for your suggestion. We have added the deviation to the obtained extraction values in Line 229 and 230. 4. Line 233: I suppose the authors refer to the concentration of the analyte (not the mass). Reply: Thanks for your suggestion. I am sorry about it caused by my careless. And we have revised the mass to the concentration of extraction yield of target analytes in Line 234.

Round 2

Reviewer 1 Report

Please check reference formatting as per journal guidelines. Journal abbreviations are not properly formatted. 

Reviewer 3 Report

The revised version of the manuscript is significantly improved and it is now acceptable for publication after minor English and editing errors are corrected.